

# Geographic distribution of free-living marine nematodes in the Clarion-Clipperton Zone: implications for future deep-sea mining scenarios

Freija Hauquier[1], Lara Macheriotou[1], Tania N. Bezerra[1], Great Egho[2], Pedro Martínez Arbizu[2], Ann Vanreusel[1]

[1] Marine Biology Research Group, Ghent University (UGent), B–9000, Gent, Belgium
[2] German Center for Marine Biodiversity Research (DZMB), Senckenberg am Meer, D–26382, Wilhelmshaven, Germany

*Correspondence to*: Freija Hauquier (freija.hauquier@ugent.be)

**Abstract.** Mining of polymetallic nodules in abyssal seafloor sediments promises to address the growing worldwide demand for metallic minerals. Given that prospective mining operations are likely to have profound impacts on deep seafloor communities, industrial investment has been accompanied by scientific involvement for the assessment of baseline conditions and provision of guidelines for environmentally sustainable mining practices.

Benthic meiofaunal communities were studied in four prospective mining areas of the Clarion-Clipperton Zone (CCZ) in the

east Pacific Ocean, arranged in a southeast-northwest fashion coinciding with the productivity gradient in the area. Additionally, samples were collected from an Area of Particular Environmental Interest (APEI-3) in the northwest of the CCZ where mining will be prohibited and which should serve as a 'source area' for the biota within the larger CCZ. Total densities in the 0–5 upper cm layer of the sediment were influenced by sedimentary characteristics, water depth and nodule density at the various sampling locations, indicating the importance of nodules for meiofaunal standing stock.

Nematodes were the most abundant meiobenthic taxon and displayed a relatively similar community composition for the different areas. Assemblages were typically dominated by a few genera (generally 2–6), accounting for 40–70 % of all individuals, as well as a large number of rare genera each contributing less than 5 % to the overall abundances. Dominant genera were widely spread within the CCZ and shared among all sampled license areas, whereas rare genera were usually limited to one. The same trend was present when looking at the species level of one of the dominant genera, *Halalaimus*,

implying that it might be mainly these rare genera and species that will be affected by changes in their habitat due to mining activities.

## 1 Introduction

As mining of mineral resources on land is increasingly burdened with logistic and geopolitical concerns (Petersen et al., 2016), humankind is now looking towards the deep seafloor (~ 4000–5000 m) as a potential source to meet the global

demand for metallic deposits. The abyssal plains of the Clarion-Clipperton Zone (CCZ) in the central eastern Pacific harbour the largest known accumulation of polymetallic nodules, rich in nickel, manganese, copper and cobalt (Halbach and Fellerer,

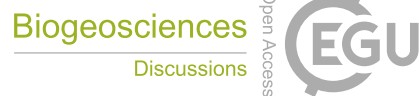

1980;Mewes et al., 2014). Their occurrence is relatively well-studied and has sparked economic interest from governments and industry. Since the CCZ falls beyond national jurisdiction, the regulatory framework for prospective nodule mining is provided by the International Seabed Authority (ISA), which has so far granted sixteen exploration licenses to interested parties (https://www.isa.org.jm/deep-seabed-minerals-contractors). One of the requirements of the ISA toward contractors is documentation of the biota in their license area and evaluation of the impact that planned mining activities will have on the environment (Vanreusel et al., 2016;Lodge et al., 2014). However, beside these baseline exploration studies, scientific knowledge on the abyssal communities of the CCZ is relatively scant, and generally of limited geographic coverage (Lambshead et al., 2003;Singh et al., 2016).

The abyssal seafloor (> 3000 m) represents the largest benthic ecosystem of the planet, encompassing over 90% of the world's oceans. Yet faunal diversity and drivers structuring spatial and temporal variability in benthic communities remain poorly understood (McClain and Schlacher, 2015;Ramirez-Llodra et al., 2010;Glover and Smith, 2003). Most deep-sea ecosystems are classified as heterotrophic, since they largely depend on photosynthetic production in overlying waters for their food requirements (Ramirez-Llodra et al., 2010;Radziejewska, 2014). Despite this dependency, which is further complicated by low sedimentation rates at these depths (Petersen et al., 2016;Gollner et al., 2017), biodiversity is high in most deep-sea habitats, especially for the smaller-sized macro- and meiofauna (Ramirez-Llodra et al., 2010). This is partially a consequence of high heterogeneity that exists as a result of the complex geological and hydrological features of the deep seafloor (Vanreusel et al., 2010), which create microscale patchiness in both abiotic and biotic features. Due to their particular role in several ecosystem functions, such as the transfer between bio- and geosphere and modulating biogeochemical cycling of carbon, nutrients and trace elements (Lessin et al., 2018), benthic faunal communities are a crucial constituent for consideration in deep-sea mining bioassessments.

Here the benthic meiofauna of the CCZ, an intermediate component between small microbes and larger-sized macro- and megafauna was investigated. This size class is mainly dominated by the phylum Nematoda, which occur in relatively high densities and biodiversity in abyssal soft sediments. Moreover, habitat heterogeneity (Vanreusel et al., 2010) and productivity regimes in overlying waters (Lins et al., 2014;Pape et al., 2013) dictate variability and patchiness in assemblages of this taxon (see also (Radziejewska, 2014)). Over the past few decades several studies have investigated meiofaunal communities, and nematodes specifically, within the CCZ (e.g., (Lambshead et al., 2003;Mahatma, 2009;Miljutin et al., 2015;Miljutina et al., 2010;Pape et al., 2017;Radziejewska, 2014;Singh et al., 2016)); however, most of them focused on one particular area. This study reports data for meiofauna across a broad latitudinal and longitudinal range within the CCZ and includes species-level information for a typical subdominant deep-sea genus of nematodes, *Halalaimus* (see (Sebastian et al., 2007;Miljutina et al., 2010;Vanreusel et al., 2010) and references therein).

The main focus of this study is the characterization and comparison of nematode communities at six sites, located in four different license areas and the Area of Particular Environmental Interest #3 (APEI-3). Moreover, we aim to link any patterns of density, diversity and/or community composition to environmental conditions, nodule densities as well as the observed latitudinal productivity gradient (Vanreusel et al., 2016). In terms of mining impacts, we specifically want to identify



nematode genera and species distributions across the wider CCZ, given that recovery of communities after mining will largely depend on the degree of connectivity between impacted and non-impacted zones. We hypothesise that widespread taxa would have an advantage in this case.

## 2 Material and methods

### 2.1 Sampling strategy

Sampling was conducted in the CCZ during the EcoResponse cruise SO239 with RV *Sonne* (Martínez Arbizu and Haeckel, 2015) in March-April 2015. Six different sites situated in four license areas and APEI-3, as established by the ISA, were visited to study the biological, geological and geochemical characteristics across a productivity gradient (Fig. 1, Table 1). License areas were those of contractors BGR (Germany), IOM (Inter-Ocean Metal consortium), DEME (GSR, Belgium) and

IFREMER (France), while APEI-3 was chosen as a no-mining reference. The German license area was further subdivided in a 'Reference area' (BGR_RA; limited future mining) and 'Prospective area' (BGR_PA; intensive future mining), leading to the following six sampling sites: APEI-3, IFREMER, GSR, IOM, BGR_RA and BGR_PA. The different sampling sites are located in an easterly fashion following a gradient in particulate organic carbon (POC) input (Vanreusel et al., 2016). Sampling depth increased in the opposite direction (Table 1). Finally, there is a latitudinal gradient in sampling locations as

well, with APEI-3 being located farther to the north, IFREMER and GSR in the middle, and IOM and BGR in the south of the CCZ.

In each area, 3–6 multicorer (MUC; 12 cores; inner diameter 94 mm) deployments were carried out to retrieve undisturbed seafloor sediment samples for analyses of meiofauna and abiotic variables. From each deployment, one or two cores were preserved for meiofauna analysis and one for measurement of environmental variables. Meiofauna cores were stored as

either bulk samples, i.e. preservation of the upper 0–5 cm in a borax-buffered formalin-seawater solution (final concentration 4–8 %), or sliced per cm down to 5 cm sediment depth for the IOM samples (each cm slice stored separately on formalin). All meiofauna, except that of the BGR areas, was separated from the sediment by density-gradient centrifugation ($3 \times 12$ min at 3000 rpm) with the colloidal silica polymer Ludox® HS-40 as a flotation medium (specific density 1.18 g cm$^{-3}$; (Heip et al., 1985;Vincx, 1996)) at the Marine Biology lab (Ghent University, Belgium). A similar protocol was adopted for the

BGR samples at the DZMB lab (Senckenberg, Germany), using a slightly different centrifugation protocol ($3 \times 6$ min at 4000 rpm) and different flotation medium (Levasil H.C. Stark 200/40 %; specific density 1.17 g cm$^{-3}$). Retention of meiofauna was in both cases achieved using a 32 µm sieve, and individuals were dyed with Rose Bengal (0.5 g l$^{-1}$) to facilitate identification. Meiofauna individuals were counted and identified to higher taxon level under a stereomicroscope ($50 \times$ magnification) using the guide of (Higgins and Thiel, 1988).

From each cm-slice (IOM) or bulk sample (all other areas), between 120 and 320 nematodes were randomly picked, transferred to anhydrous glycerol (Seinhorst, 1959;De Grisse, 1969) and mounted on slides. Genus-level identification was performed under a Leica DMLS compound microscope ($1000 \times$ magnification) according to the pictorial keys of (Platt and



Warwick, 1983;Platt and Warwick, 1988;Warwick et al., 1998), and the information contained in the NeMYS database (Bezerra et al., 2018). Undamaged mounted adult specimens of the genus *Halalaimus* were further identified to species level, with the aid of the pictorial key of (Platt and Warwick, 1983) and other relevant species descriptions (Bussau, 1993). Individuals were carefully assessed for their morphometric characteristics, vouchered by means of detailed photographs, and

5 measured to calculate de Man's ratios (Fortuner, 1990).

Cores for measurement of abiotic variables were sliced in two parts (0–1 cm and 1–5 cm) and stored at -20 °C until further analysis at the UGent Marine Biology Research Group. A 1 ml subsample from each core was obtained and stored at –80 °C for pigment analysis which was carried out at the Max Planck Institute for Marine Microbiology (MPI Bremen). Several sediment parameters were obtained from each slice, but since pigments were only measured in surface sediments,

environmental variables used in later analyses are those for the 0–1 cm slice. Grain size distribution was determined by laser diffraction (Malvern Mastersizer hydro 2000 G, size range 0.02–2000 µm) and classified according to (Wentworth, 1922). For this study, clay (sediment particles < 4 µm) and silt (4–63 µm) size fractions were considered in further analyses. Weight percentages of total organic carbon (TOC) and nitrogen (TN) were measured by combustion of freeze-dried samples using a Flash 2000 NC Sediment Analyser (protocol through Interscience B.V., Breda, The Netherlands). Pigment analysis included

determination of concentrations of chlorophyll a and its degradation products, phaeopigments, the sum of which constitutes the Chloroplastic Pigment Equivalents (CPE, µg ml$^{-1}$). Both chlorophyll and phaeopigments were extracted by means of 90 % acetone, and their concentrations were determined through fluorometry (Trilogy® Laboratory Fluorometer, Turner Designs) according to manufacturer protocol. Finally, depth of the different sampling locations (a proxy for other – unmeasured – environmental variables that tend to vary with water depth), and approximate nodule density (in kg m$^{-2}$) as

determined from nodule counts and quantification of box core surfaces taken in the same area (50 × 50 cm; (Martínez Arbizu and Haeckel, 2015)) (Table 1), were included in later statistical analyses.

### 2.2 Statistical analyses

Total meiofauna and nematode densities were standardized to number of individuals per 10 cm² prior to further analysis. A one-way analysis of variance (ANOVA, factor 'area' with 6 levels) was used to check for differences in densities among the

25 sampling areas, after assumption checking in R (R Core Team, 2013). The relationship between densities and environmental variables was investigated through linear regression models (step-wise selection procedure, Akaike Information Criterion AIC) in R.

Differences in environmental conditions as well as nematode genus communities between the different areas were assessed by means of permutational analysis of variance (PERMANOVA) in PRIMER v6 (Clarke and Gorley, 2006) with the

30 PERMANOVA+ add-on (Anderson et al., 2008). The design included one fixed factor (area) and resulting P-values were based on 9999 permutations (unrestricted permutation of raw data; type III sum of squares). True permutational P-values *P*(perm) were interpreted when the number of unique permutations exceeded 100; alternatively, Monte Carlo P-values *P*(MC) were used. A PERMDISP test was carried out to assess homogeneity of dispersions in the multivariate space



(distances to centroids; P-value by permutation of least-squares residuals). All environmental variables were standardised to zero mean and unit variance prior to analyses and (dis)similarity quantified using Euclidean distance matrices. Both silt and clay content were log-transformed to account for skewness in the data (assessed by draftsman plots). Differences between areas were then visualized by principal components analysis (PCA). In a similar fashion, differences in nematode

communities of the different areas were visualized through non-metric multidimensional scaling plots (nMDS) based on Bray-Curtis similarities. Nematode genus counts were first standardized to relative abundances (to account for differences in the number of identified specimens), and then square-root transformed (Hellinger transformation) to reduce the influence of highly dominant genera. Since the number of nematodes that could be identified in each replicate sample varied among areas, absolute genus counts were rarefied to the lowest number of identified individuals (98) with the 'rrarefy' function in

the package 'vegan' (Oksanen et al., 2017) in R. The use of rarefied counts yielded similar results in all analyses and are therefore not reported unless specified otherwise. Multivariate relationships between nematode community assemblages and environmental variables were investigated through the distance-based linear model (DISTLM) procedure in PERMANOVA+, the results of which were visualized in distance-based redundancy analysis (dbRDA) plots. The average number of specimens identified, the number of genera (observed and expected in a sample of 98 individuals – to account for

differences in the number of individuals identified, see earlier), Hill's $N_1$ (Hill, 1973) (the true numbers' equivalent of the Shannon-Wiener entropy (Jost, 2006)) and Pielou's evenness J' were calculated for each area using PRIMER v6. Differences in diversity and evenness indices between sampling areas were assessed by means of one-way ANOVA in R (factor 'area' with 6 levels).

## 3 Results

Apart from BGR_RA and IOM, all areas within the CCZ showed substantial differences from each other with respect to environmental characteristics (significant PERMANOVA main test result, significant pairwise differences; Table 2; Fig. 2). The APEI-3, located farther to the north, had a finer sediment composition (highest clay fraction) and lower levels of pigments and TOC (Table 1; Fig. 2). A northwest-southeast trend in organic matter content was visible in the sediments, as CPE and TOC increased from APEI-3 to BGR. Other variables that differed notably between sampling sites were depth

(increasing from east to west), and nodule density, with highest nodule densities in the sites sampled in GSR, IFREMER and BGR_PA license areas (Table 1).

Meiofaunal densities ranged between ~50 and 550 ind. 10 cm⁻², with significantly lower numbers in APEI-3 compared to the other areas (Table 3). Nematodes were clearly dominant (~86–91 % of total communities), followed by copepods and nauplii (~7.5–10.5 %) while other taxa occurred in very low abundances (max. 3 %). Multiple linear regression of nematode

densities in function of the environmental variables yielded a model containing variables clay, depth and nodule density ($F_{3,19} = 24.33$; $P < 0.001$; $R^2_{adj} = 0.76$; after removal of one outlier), all negatively influencing densities. Nematode densities





were clearly lowest in the APEI-3 (Table 3; one-way ANOVA: $F_{5,18} = 11.3$; $P < 0.001$), characterized by a low amount of nodules, the finest sediment, and relatively deep location (Table 1), and highest in BGR and IOM.

Nematode genus composition was significantly different between areas (significant PERMANOVA main test, $P < 0.05$; PERMDISP not significant: $P = 0.3661$), although this effect could not be investigated in further detail due to the low number of unique permutations (Table 4). The same pattern arose (PERMANOVA main test $P = 0.0002$; results not shown) when the analyses were repeated with genera grouped into families with the difference that pairwise comparisons of each of the areas IOM, IFREMER and APEI with BGR_PA were significant despite low permutation numbers (PERMANOVA pairwise test; results not shown). Only when areas were grouped in a latitudinal fashion (APEI = northwest CCZ; IFREMER and GSR = middle CCZ; IOM and BGR = southeast CCZ) did all pairwise comparisons give significant differences (all $P < 0.02$; results not shown). Overall, average within-area similarity ranged between 50 and 57 %, while between-area similarity was only slightly less, ranging between 45–57 %.

No clear relationship between community assemblages and environmental parameters could be discerned. The DISTLM procedure pointed towards 'depth' and 'nodule density' as the main variables explaining the community variation (both marginal and sequential P-values significant), but together they only accounted for 19.4 % of the variation (results not shown).

In total, 156 different genera belonging to 34 families were identified across all areas with average number of genera per area ranging between 32 and 50 (Table 5). The expected number of genera in a sample of 98 individuals (taking into account differences in the number of identified individuals per sample) was comparable for the different areas (one-way ANOVA: $P = 0.43$), yet slightly (but insignificantly) lower for the BGR_PA stations (Table 5). Hill's index $N_1$, which takes genus richness as well as evenness into account, was highest for GSR and IOM (one-way ANOVA: $P = 0.58$). Based on Pielou's J', evenness was roughly similar in most areas, yet significantly lower in BGR stations compared to GSR and APEI-3 (one-way ANOVA: $F_{5,13} = 6.31$; $P = 0.0035$). Communities were typically dominated by a few genera belonging to the families Monhysteridae, Xyalidae, Chromadoridae and Oxystominidae, which occurred in high relative abundances (> 5 %) (Table 5, 6). Genera *Acantholaimus*, *Halalaimus* and *Monhystrella* displayed especially high relative abundances in one or multiple sampling areas (Table 6). Whereas *Monhystrella* was the most abundant genus across all areas (except for BGR_PA), the contribution of *Halalaimus* was largest in the APEI-3, while *Acantholaimus* became more important towards the southeast of the CCZ (BGR). *Halalaimus* showed a significant negative relationship with the amount of silt in the sediments ($R^2_{adj} = 0.633$; $P < 0.001$). In the case of *Acantholaimus*, absolute counts were negatively correlated with depth ($R^2_{adj} = 0.495$; $P < 0.001$), while relative abundance was positively associated with CPE values ($R^2_{adj} = 0.499$; $P < 0.001$), hence its higher contribution in BGR sediments (Table 1).

Notably, most of the genera occurring in high relative abundances were shared among the different sites. Of the 156 genera identified, roughly 15 % (corresponding to 23 genera) were shared among all sampling sites while nearly half (~44 %; 68 genera; or ~46 % and 72 genera when BGR is considered as one) were restricted to a single site (so-called 'unique' genera; Fig. 3 left panel; Table 5). Unique genera contributed only a minor fraction to total nematode densities (Table 5), and can





thus be considered 'rare'. Furthermore, within-area variation in genus composition was also substantial, with more than half of the genera (~51 %) on average being unique within a single replicate sample. In the IOM area, nearly 60 % of the genera were found only once, while this number was much lower (43 %) in the IFREMER license area. Relatively more genera were shared between the mid (GSR, IFREMER) and southeast (IOM, BGR) latitudinal range of the CCZ, than between

either of those and the APEI area in the northwest (Fig. 3 right panel).

A total of 143 individuals of the genus *Halalaimus* could be assigned to species level and were classified into 24 different morphospecies (Table 7). As was the case for nematode genera, a limited number of morphospecies was shared among all sampling areas (5 species), while several species had restricted distributions, occurring in just one or two areas (Table 7, Fig. 3 right panel). The highest species diversity (18) of *Halalaimus*, which was also accompanied by the highest relative

abundances, was found within the APEI-3 (Fig. 3).

## 4 Discussion

The importance of meiofauna in deep-sea abyssal sediments has been demonstrated repeatedly (e.g., (Wei et al., 2010)). While biomass, abundance and size of organisms of different size classes (mega-, macro- and meiofauna) all show a negative trend with increasing water depth, this decrease is less pronounced for the meiofauna (Wei et al., 2010;Rex et al., 2006)

hence their numerical dominance in abyssal plains. Nematode densities in this study were comparable to those reported for other abyssal and CCZ seafloor sediments (e.g., (Pape et al., 2017;Veit-Köhler et al., 2011;Lins et al., 2014)), except for the rather low numbers in the APEI-3 area. Meiofaunal abundance, and that of nematodes in particular, is known to vary with the available food concentrations in the sediment (generally assessed by means of pigment concentrations), as well as with other environmental characteristics (Lins et al., 2014;Moens et al., 2013). Similarly, several environmental variables in this

study were highly correlated with faunal densities, most notably nodule density, water depth and clay content. Contrary to what has been observed for other deep-sea regions worldwide (e.g., (Ramirez-Llodra et al., 2010;Lins et al., 2014;Pape et al., 2013)), the importance of phytodetritus-derived food at the seafloor (approximated by CPE and TOC values; Table 1) was not reflected directly in nematode standing stock and community composition, although their values varied considerably among the different areas of the CCZ (Table 1, Fig. 2). Only the relative importance of the genus *Acantholaimus* showed a

positive relationship with CPE concentrations among the various locations. The productivity gradient (POC flux; (Vanreusel et al., 2016)) in the area was therefore reflected mainly in the sediment abiotic variables and only to a lesser extent in the faunal parameters. Given that the entire CCZ is classified as oligotrophic (Volz et al., 2018), concentrations of phytodetritus-derived pigments might have been too low to result in any obvious explanatory relationship with faunal variables.

One of the parameters that did influence faunal densities was nodule density of the different areas. Previous work on benthic

fauna of different size classes showed the importance of nodules in structuring communities (Miljutina et al., 2010;Vanreusel et al., 2016;Radziejewska, 2014). Judging from the results of this study, high nodule densities were associated with a lower meiofaunal standing stock (Tables 1, 3). This may seem rather logical since an increase in nodule density leads to a decrease





in the volume of fine-grained sediments which constitute the biotope of meiofauna and nematodes; however, it might also be the result of sediment volume differences due to the presence of nodules in the sampling cores. Nevertheless, a similar negative correlation between nodule density and nematode counts was observed at a smaller scale (Miljutina et al., 2010), where nodule-bearing sediments in the IFREMER license area displayed lower average nematode abundances than nodule-

free sites within the same area. Similarly in our study, high-nodule sediments of GSR, IFREMER and BGR_PA (see Table 1) had lower nematode densities than low-nodule IOM and BGR_RA. The APEI-3 was exceptional as it was characterized by low nematode abundances despite low nodule densities. This area differed significantly from the rest in additional aspects such as lower CPE and TOC values and finer sediment, with an increase in the clay fraction (< 4 µm; Table 1; Fig. 2). This was corroborated by a more detailed assessment of CCZ sediments which showed that biogeochemical features of the APEI-

3 area differed considerably from other areas (Volz et al., 2018). Therefore, a different sedimentation rate resulting in finer sediments with lower POC input (Volz et al., 2018) may be the justification for the deprived nematode numbers at the APEI-3 stations rather than the effect of nodule densities.

As nodules are confined to the upper sediment layer (Petersen et al., 2016; Radziejewska, 2014), where the majority of the meiofauna also resides, their removal through mining activities will induce drastic changes to these small-sized biota (Thiel,

2001; Miljutin et al., 2015). Inevitably, the ploughing and removal of surface sediments associated with the mining process will lead to an initial decrease in densities (Shirayama et al., 2001; Radziejewska, 2014; Jones et al., 2017). Yet the ratio of nodule presence to bare sediment availability is not the only structuring factor of nematode communities, based on the results of this study. Rather, a complex interplay of several factors (here: particle size, water depth and nodule density) ultimately defines nematode densities. Other, more indirect, impacts of deep-sea mining activities (see (Gollner et al.,

2017; Thiel, 2001; Radziejewska, 2014) and references therein) might therefore have an even larger impact on meiofauna numbers through their interference with the sediment biogeochemistry. Specifically, sediment blanketing and displacement which will accompany large-scale mining operations will likely impose considerable disturbances (Radziejewska, 2014). As shown before for the drilling industry, discharge of deep-sea muds can seriously impact meiofaunal communities, both in terms of abundance as well as community composition (Netto et al., 2009). In addition to chemical effects, physical

alteration of the habitat (i.e. change in sediment granulometry and burying of organisms) was primarily responsible for the lack of recovery in communities one year after the disturbance took place (Netto et al., 2009). Also *in situ* disposal and deposition of nodule debris onto disturbed seafloor sediments can result in altered behaviour of small-sized meiofauna communities as demonstrated in a small-scale experiment in the Peru Basin (characterized by similar presence of abyssal nodules; see (Mevenkamp et al., 2018)). Over an incubation period of 11 days, several meiofaunal taxa, including

nematodes, changed their vertical position in the sediment after the addition of a top layer with crushed nodule particles (Mevenkamp et al., 2018). Finally, sediment compaction by vehicle tracks might also change sediment characteristics (e.g., porewater geochemistry), resulting in the inability of some meiofauna to penetrate these compacted sediments (i.e. explanation why densities were still deprived after 26 years in (Miljutin et al., 2011)). Moreover, the refractory carbon pool may be removed by mining and prevent recolonization.

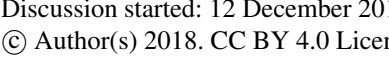


It has been shown by various studies within the CCZ that nematode diversity at genus and morphospecies level is generally high at a local scale (see (Radziejewska, 2014) and references therein). Additionally, dominance of genera is usually limited and therefore evenness high (Radziejewska, 2014). With a total of 156 genera reported in this study, genus diversity was indeed substantial on a regional scale and only few genera occurred in higher numbers (Tables 5, 6). Overall, diversity was

similar for the different areas, but evenness was lower in the BGR stations (Table 5). This could be attributed to the presence of two highly dominant genera, *Acantholaimus* (> 17 %) and *Monhystrella* (> 21 %; Table 6), as compared to only *Monhystrella* in the other areas (> 25 %; Table 6). Nematode communities generally consisted of typical deep-sea families and genera such as *Monhystrella* (and Monhysteridae in general), *Acantholaimus* and *Halalaimus*, all of which have been reported from previous CCZ samplings (Singh et al., 2016;Radziejewska, 2014;Miljutin et al., 2015;Miljutin et al.,

2011;Miljutina et al., 2010;Pape et al., 2017;Lambshead et al., 2003) (see Table 6) and other abyssal areas worldwide (Lins et al., 2014;Vanreusel et al., 2010;Singh et al., 2016). Differences between results described here and previous CCZ studies mainly lie in their relative abundances. For instance, higher contributions of the genus *Theristus* (ranging between 3.6 and 13 %) were reported in (Miljutin et al., 2015;Miljutina et al., 2010) for the French license area, while here it was subdominant across the CCZ (max. 4.4 %; Table 6). As was already observed before for the CCZ region, faunal differences among

samples in nodule areas are typically driven by a high contribution of 'rare' species (defined as species with few individuals and/or a restricted geographical distribution; see definition reported in (Cao et al., 2001) and references therein) (Gollner et al., 2017). Also in this study, a high relative proportion of rare genera (~44 % occurring in only 1 site) and species (~63 % if BGR_PA and BGR_RA are considered as one area) were recovered in the different sampling areas. Moreover, even within a single area, roughly 50 % of the genera were found in only one of the replicates (nearly 60 % for the IOM area) with low

abundances (generally only 1 or 2 individuals). Similar patterns could be observed for the morphospecies of *Halalaimus*. A total of 24 morphospecies of *Halalaimus* were identified in this study for the entire CCZ. This number is higher than the 11 and 13 morphotypes reported by (Lambshead et al., 2003) and (Miljutina et al., 2010), respectively, but comparable to 25 morphotypes reported for the Peru Basin under relatively similar conditions in terms of depth and nodule presence (Bussau, 1993). Some of the morphospecies were also described in the study by (Pape et al., 2017) for the GSR license area,

indicating their consistent presence over time. As for the genera, most of these species were found in only one or two areas, in low numbers (although not all *Halalaimus* individuals could be assigned to species level).

Our findings in terms of genus and species distribution have important implications for future mining scenarios and suggest that in the event of large-scale mining efforts, rare genera and/or species are likely to disappear completely from targeted areas. It is thus crucial to know what the impact of such a loss of species will be in a deep-sea context, especially in terms of

ecosystem functioning. It may be that these effects are obscured at first, but become more apparent over time (as demonstrated in various habitats; see (Gazcon et al., 2015) for examples), depending on the redundancy of the system. However, knowledge on the latter is particularly limited in the abyss (Radziejewska, 2014) and future studies should therefore aim to elucidate the particular roles of rare deep-sea species. Additionally, the effect of rare species loss could be transferred to other levels of the trophic chain, and, as was shown in a rocky intertidal study (Bracken and Low, 2012), have

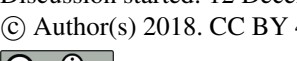



a proportionally larger impact on the consumers than an equal loss of dominant species would have. Again, little is known about the importance and function of rare taxa in nematode communities. Finally, in terms of genetic diversity, loss of species in mined areas might result in diminished population exchange and genetic variation in the areas within and around mining sites, as was shown for massive sulphides (Boschen et al., 2016;Gollner et al., 2017). Despite the initial risk for a

5 substantial loss of rare taxa after mining disturbance, an on-board experiment by (Gallucci et al., 2008) showed that species colonising newly exposed sediments were mostly rare or undetectable in control treatments. Based on these findings one can conclude that disturbance is a prerequisite for some species to be able to persist in deep-sea sediments, as these may benefit from the availability of space left behind by the removal of dominant taxa. If this can be generalised to all rare taxa, regardless of the circumstances, it means that in the case of large-scale mining, there might actually be an increase in density

of rare nematode species rather than a decrease.

True rarity of taxa is only one of the factors that will influence the outcome of potential recolonization events after mining operations. Additionally, recovery of communities will depend to a large extent on the provision of food resources and the presence of a pool of colonisers in the vicinity of the disturbed areas (Radziejewska, 2014), both within single license areas (important for local dynamics) as well as across the larger CCZ (important for regional species dynamics). Low connectivity

will result in slow recovery potential of species (Gollner et al., 2017) as unmined sites within the license areas or the envisioned source areas (APEIs) will not be able to replenish the standing stocks of lost taxa. Again, given that many genera and *Halalaimus* species occurred in low numbers and with a limited geographic spread, source areas might fall short of their role for population replenishment. The number of rare taxa is a potential overestimation and would likely decrease with increasing sampling effort; however, the observed rarity is only partially the result of undersampling. In the context of deep-

sea mining, it will therefore be of utmost importance to monitor post-impact communities to identify recovery over space and time. Results from previous work in different benthic environments demonstrated that (re-)colonization by nematodes after disturbance events can take up to several decades (e.g., after iceberg scouring in Antarctic waters; (Lee et al., 2001)). Within the CCZ specifically, the recovery of nematode communities in a disturbance track in the IFREMER license area was found to be rather slow (Miljutin et al., 2011). Whereas densities are generally known to recover faster to pre-disturbance

conditions (see (Jones et al., 2017;Radziejewska, 2014) and references therein), this is not necessarily true for community composition (Miljutin et al., 2011). Although some genera have been shown to actively colonise disturbed sediments (Gallucci et al., 2008), most genera are considered to have limited dispersal capacities (Derycke et al., 2013;Moens et al., 2013). Especially at larger geographic scales, it seems that at least some species are limited in their distribution (Hauquier et al., 2017;Derycke et al., 2013). In combination with the lack of a pelagic larval stage in nematodes, this further reduces their

capacity for quick community recovery after disturbance. Nevertheless, passive transport of nematodes by bottom currents eroding the topmost sediment layers has been documented (Boeckner et al., 2009;Thiel, 2001) although current speeds near the seafloor in the CCZ are generally rather low (1–9 cm s$^{-1}$; (Mewes et al., 2014;Radziejewska, 2014)), casting doubt on their effectiveness for erosion and long-distance transportation of organisms and larvae.

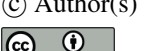


Even when nematodes are able to (re-)colonise disturbed areas, the outcomes are generally very different due to the different environmental settings and processes acting on them. This effectively hampers our ability to predict responses of nematode assemblages to large-scale mining events. The lack of a consensus is demonstrated by the differing outcomes of several artificial disturbance studies carried out since the 1970s in a nodule context (see (Jones et al., 2017) and references therein).

For instance, in the IOM-BIE experiment, one particular group of nematodes (Desmoscolecids) was responsible for a peak in post-impact abundances (Radziejewska, 2002), while in another CCZ study (Miljutin et al., 2011), this was attributed to the genera *Thalassomonhystera* and *Oncholaimus*. Another drawback for future predictions is the fact that most of these studies looked at recovery processes at relatively small scales, and are therefore not necessarily representative of the large-scale impact of mining operations at hand (Gollner et al., 2017). As far as can be predicted at this moment, the extraction of the

nodules from the seabed will impose a disturbance to all biota at a scale that ranges from 10s to 1000s of km² (Glover and Smith, 2003). Although disturbance events acting at multiple spatial scales (McClain and Schlacher, 2015) are not uncommon in the deep sea, the consequences thereof remain difficult to predict. The effects of the sediment plumes associated with mining especially are a major challenge, given that there is a lot of uncertainty on the spatial extent of this type of impact (Gollner et al., 2017;Radziejewska, 2014). Previous experiments mimicking mining effects and subsequent

plume dispersal on a smaller scale have indicated that traveling distances of the bottom plumes ranged from a few hundreds of metres to several kilometres away (Peukert et al., 2018;Sharma et al., 2001).

As was already stated in previous works (e.g., (Vanreusel et al., 2016;Gollner et al., 2017)), in order to preserve current community structure, unmined areas should be comparable to provisioned mining sites (Boetius and Haeckel, 2018) and function as recruitment sources after mining takes place; the latter, however, remains to be tested (Kaiser et al., 2017). This

study therefore also focused on the APEI-3, in the northwest of the CCZ. While these areas were originally selected based on environmental proxies, our data shows that the APEI-3 at least, exhibits distinct biotic and abiotic characteristics compared to the other sampled areas (see also (Volz et al., 2018)) as densities and community composition were significantly different (i.e. low abundances despite low nodule densities and a higher contribution and species number of *Halalaimus*; Tables 1, 3, 6; Fig. 3). While information for the other APEIs is scarce, especially in comparison to the license areas, data from this study

suggest that the APEI-3 is ill-suited as a representative source area for the recovery of the mined communities. Additionally, given the large geographic distance between the APEI-3 and the license areas, exchange of individuals, hence genetic material, among remaining populations after mining may be hampered. In the case of nematodes, which are known to have limited dispersal capacity (see earlier), a network of APEIs that are randomly spread across the CCZ might have been more beneficial (as originally planned; see (Kaiser et al., 2017) and references therein). Therefore, as was suggested by (Vanreusel

et al., 2016) we strongly advocate for the incorporation of no-mining sites within each of the license areas, in order to buffer the inevitable loss of biodiversity.

**Data availability**

Datasets supporting this manuscript can be accessed online through the PANGAEA database at
https://doi.org/10.1594/PANGAEA.873272 and https://doi.org/10.1594/PANGAEA.873274.

**Author contribution**

LM, AV and PM participated in expedition SO239 and collected the samples. FH, LM, PM and AV discussed sampling
strategy and analyses to be performed. FH, TN and GE analysed the samples and performed the identifications. FH and AV
executed statistical data analyses, with further contributions of LM and PM. All authors contributed to the writing of this
manuscript.

**Competing interests**

The authors declare that they have no conflict of interest.

**Acknowledgements**

The authors wish to thank the captain, crew, chief scientist and scientific community of expedition SO239 onboard RV
*Sonne* for their help and support in sample acquisition. Dr. Felix Janssen from MPI Bremen/AWI and co-workers are greatly
acknowledged for the analyses of pigment data. Bart Beuselinck and Niels Viaene from UGent are thanked for their
assistance in the analyses of abiotic parameters and extraction of meiofauna samples. Annick Van Kenhove and Guy De
Smet from UGent helped in picking of nematodes and making of slides. The SO239 cruise with the RV *Sonne* was financed
by the German Federal Ministry of Education and Research BMBF (grant n° 03F0707A-G) as part of the JPI-Oceans pilot
action MiningImpact – "Ecological Aspects of Deep-Sea Mining". Additional funding was available through the Department
of Economy, Science & Innovation of Flanders under the framework of JPI Oceans (grant n° 1242114N). Data are available
through the online PANGAEA portal.

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





**Tables**

**Table 1. Upper panel: sampling date, geographic location, depth, and approximate nodule coverage (based on information in (Vanreusel et al., 2016)) for the different license areas and APEI-3.** Note that geographic locations are an approximation since each area encompasses a number of replicates that are located a few m's apart. **Lower panel: average environmental parameters and standard deviation in the 0–1 cm layer.** CPE = Chloroplastic Pigment Equivalents, TN = Total Nitrogen, TOC = Total Organic Carbon, clay = sediment fraction with grain size < 4 µm, silt = sediment fraction with grain size between 4 and 63 µm.

|  | Date | Latitude | longitude | depth (m) | nodule density (kg m⁻²) |
|---|---|---|---|---|---|
| **APEI-3** | 23/04/2015 | 18°47.35' N | 128°21.26' W | 4839.1 | $3.65 \pm 2.34$ |
| **IFREMER** | 16/04/2015 | 14°02.60' N | 130°07.82' W | 4964.6 | $19.97 \pm 4.05$ |
| **GSR** | 9/04/2015 | 13°51.28' N | 123°14.69' W | 4511.8 | $26.15 \pm 1.10$ |
| **IOM** | 2/04/2015 | 11°04.63' N | 119°39.60' W | 4434.5 | $0.70 \pm 0.44$ |
| **BGR_RA** | 30/03/2015 | 11°47.88' N | 117°30.62' W | 4342.2 | $3.20 \pm 4.23$ |
| **BGR_PA** | 24/03/2015 | 11°51.06' N | 117°03.46' W | 4123.9 | $21.80 \pm 1.15$ |

|  | CPE (µg ml⁻¹) | TN (weight %) | TOC (weight %) | clay (%) | silt (%) |
|---|---|---|---|---|---|
| **APEI-3** | $0.06 \pm 0.00$ | $0.10 \pm 0.00$ | $0.29 \pm 0.02$ | $35.48 \pm 5.40$ | $61.63 \pm 4.91$ |
| **IFREMER** | $0.08 \pm 0.02$ | $0.12 \pm 0.03$ | $0.40 \pm 0.07$ | $15.41 \pm 1.77$ | $71.89 \pm 3.80$ |
| **GSR** | $0.11 \pm 0.05$ | $0.16 \pm 0.07$ | $0.47 \pm 0.11$ | $15.64 \pm 1.66$ | $70.89 \pm 2.42$ |
| **IOM** | $0.17 \pm 0.03$ | $0.12 \pm 0.01$ | $0.53 \pm 0.12$ | $10.74 \pm 0.47$ | $73.39 \pm 0.89$ |
| **BGR_RA** | $0.20 \pm 0.09$ | $0.10 \pm 0.02$ | $0.43 \pm 0.12$ | $11.21 \pm 0.89$ | $72.90 \pm 1.27$ |
| **BGR_PA** | $0.28 \pm 0.11$ | $0.12 \pm 0.01$ | $0.58 \pm 0.08$ | $12.21 \pm 0.65$ | $70.31 \pm 3.01$ |





**Table 2. PERMANOVA main and pairwise test results based on environmental variables.** PERMANOVA design with one fixed factor (area; 6 levels). P-values calculated by permutation (9999 permutations) at 5 % significance level. df = degrees of freedom, SS = sum of squares, MS = mean squares, Pseudo-F/t = effect size, $P$(perm)/$P$(MC) = permutational/Monte Carlo P-value, perms = number of unique permutations. Asterisks indicate significant P-values: * < 0.05, ** < 0.01.

**PERMANOVA**

**main test**

| Source of variation | df | SS | MS | Pseudo-F | $P$(perm) | perms |
|---|---|---|---|---|---|---|
| Area | 5 | 106.660 | 21.332 | 14.338 | 0.0001** | 9929 |
| Residual | 13 | 19.341 | 1.4878 | | | |
| Total | 18 | 126 | | | | |

**pairwise test**

| Groups | t | $P$(MC) | perms |
|---|---|---|---|
| BGR_PA vs. BGR_RA | 3.2005 | 0.0105* | 10 |
| BGR_PA vs. IOM | 3.4114 | 0.006** | 10 |
| BGR_PA vs. GSR | 3.7457 | 0.0056** | 10 |
| BGR_PA vs. IFREMER | 4.7012 | 0.001** | 35 |
| BGR_PA vs. APEI-3 | 6.5251 | 0.0006** | 10 |
| BGR_RA vs. IOM | 1.8302 | 0.1024 | 10 |
| BGR_RA vs. GSR | 3.1678 | 0.0115* | 10 |
| BGR_RA vs. IFREMER | 3.2360 | 0.0054** | 35 |
| BGR_RA vs. APEI-3 | 3.5571 | 0.0061** | 10 |
| IOM vs. GSR | 3.3624 | 0.0053** | 10 |
| IOM vs. IFREMER | 3.2318 | 0.003** | 35 |
| IOM vs. APEI-3 | 4.9877 | 0.003** | 10 |
| GSR vs. IFREMER | 2.0347 | 0.0328* | 35 |
| GSR vs. APEI-3 | 5.1015 | 0.0023** | 10 |
| IFREMER vs. APEI-3 | 4.1434 | 0.0021** | 35 |





**Table 3. Average meiofauna densities (individuals 10 cm⁻²) and standard deviation for the different CCZ areas.** Copepoda includes both adults and nauplius larvae. Other taxa = meiofaunal taxa excluding nematodes and copepods.

|          | Nematoda         | Copepoda       | Other taxa     | Meiofauna         |
|----------|------------------|----------------|----------------|-------------------|
| **APEI-3**   | 45.82 ± 6.29     | 5.71 ± 2.02    | 1.71 ± 0.63    | 53.24 ± 8.86      |
| **IFREMER**  | 162.99 ± 57.57   | 14.54 ± 4.53   | 4.62 ± 1.62    | 182.15 ± 62.42    |
| **GSR**      | 186.31 ± 40.35   | 21.78 ± 8.81   | 3.40 ± 0.73    | 211.49 ± 40.36    |
| **IOM**      | 504.58 ± 237.70  | 35.02 ± 14.42  | 13.11 ± 8.27   | 552.71 ± 260.32   |
| **BGR_RA**   | 433.77 ± 86.00   | 37.36 ± 9.35   | 12.72 ± 3.56   | 483.84 ± 95.18    |
| **BGR_PA**   | 336.43 ± 39.89   | 28.93 ± 6.47   | 6.48 ± 0.31    | 371.84 ± 33.11    |



**Table 4. PERMANOVA main and pairwise test results for nematode genus composition.** PERMANOVA design with one fixed factor (area; 6 levels). P-values calculated by permutation (9999 permutations) at 5 % significance level. df = degrees of freedom, SS = sum of squares, MS = mean squares, Pseudo-F/t = effect size, $P$(perm)/$P$(MC) = permutational/Monte Carlo P-value, perms = number of unique permutations. Asterisks indicate significant P-values.

**PERMANOVA**

**main test**

| Source of variation | df | SS | MS | Pseudo-F | $P$(perm) | perms |
|---|---|---|---|---|---|---|
| Area | 5 | 7406.1 | 1481.2 | 1.6583 | 0.0002* | 9773 |
| Residual | 13 | 11612 | 893.22 | | | |
| Total | 18 | 19018 | | | | |

**pairwise test**

| Groups | t | $P$(MC) | perms |
|---|---|---|---|
| BGR_PA vs. BGR_RA | 1.1242 | 0.3145 | 10 |
| BGR_PA vs. IOM | 1.5766 | 0.0833 | 10 |
| BGR_PA vs. GSR | 1.3795 | 0.1546 | 10 |
| BGR_PA vs. IFREMER | 1.6038 | 0.0571 | 35 |
| BGR_PA vs. APEI-3 | 1.4471 | 0.1272 | 10 |
| BGR_RA vs. IOM | 1.3324 | 0.1842 | 10 |
| BGR_RA vs. GSR | 1.2396 | 0.2299 | 10 |
| BGR_RA vs. IFREMER | 1.5702 | 0.0662 | 35 |
| BGR_RA vs. APEI-3 | 1.3551 | 0.1599 | 10 |
| IOM vs. GSR | 1.0685 | 0.377 | 10 |
| IOM vs. IFREMER | 1.0012 | 0.4303 | 35 |
| IOM vs. APEI-3 | 1.1271 | 0.3155 | 10 |
| GSR vs. IFREMER | 0.9997 | 0.4364 | 35 |
| GSR vs. APEI-3 | 1.29 | 0.1968 | 10 |
| IFREMER vs. APEI-3 | 1.1718 | 0.2606 | 35 |



**Table 5. Top:** Nematode genus diversity (averaged over samples per site ± standard deviation) in the different CCZ areas. $EG$(98) = expected number of genera in a sample of 98 individuals, $N_1$ = Hill's index, J' = Pielou's evenness. **Bottom:** Number of most abundant genera (averaged over samples per site ± standard deviation) occurring with > 5 % in at least one replicate, and the fraction these represent in terms of nematode diversity (% of total genera) and density (% of total density). Additionally, the total number of unique genera per site (i.e. summed for all replicates) is given, together with their combined contribution to total nematode density for each site.

| | N° identified | N° genera | $EG$(98) | $N_1$ | J' |
|---|---|---|---|---|---|
| **APEI-3** | 109.00 ± 6.00 | 32.33 ± 4.51 | 30.75 ± 3.91 | 17.72 ± 3.82 | 0.82 ± 0.04 |
| **IFREMER** | 104.25 ± 4.35 | 33.50 ± 1.29 | 32.46 ± 0.86 | 16.47 ± 1.38 | 0.80 ± 0.02 |
| **GSR** | 106.33 ± 5.03 | 36.00 ± 5.57 | 34.35 ± 4.60 | 19.08 ± 3.41 | 0.82 ± 0.02 |
| **IOM** | 169.00 ± 68.79 | 38.67 ± 5.86 | 30.88 ± 3.58 | 18.92 ± 1.27 | 0.81 ± 0.04 |
| **BGR_RA** | 265.67 ± 65.04 | 50.00 ± 6.08 | 30.97 ± 1.23 | 18.01 ± 3.84 | 0.73 ± 0.04 |
| **BGR_PA** | 220.33 ± 41.36 | 42.33 ± 1.53 | 28.32 ± 0.93 | 15.72 ± 0.92 | 0.74 ± 0.02 |

| | N° dominant genera (> 5%) | % of total genera | % of total density | N° unique genera/site | % of total density/site |
|---|---|---|---|---|---|
| **APEI-3** | 3.67 ± 1.53 | 11.64 ± 5.85 | 49.34 ± 11.8 | 11 | 4.28 |
| **IFREMER** | 3.00 ± 0.82 | 8.99 ± 2.65 | 48.12 ± 5.40 | 9 | 2.40 |
| **GSR** | 2.67 ± 1.15 | 7.45 ± 3.04 | 42.49 ± 8.09 | 8 | 3.76 |
| **IOM** | 3.33 ± 0.58 | 8.93 ± 3.10 | 44.65 ± 1.25 | 14 | 4.34 |
| **BGR_RA** | 2.67 ± 0.58 | 5.44 ± 1.60 | 48.92 ± 7.07 | 19 | 3.51 |
| **BGR_PA** | 4.33 ± 1.15 | 10.20 ± 2.51 | 60.48 ± 7.96 | 7 | 1.82 |





**Table 6. Overview of the most abundant genera (top;** rel. abund. > 5 % in at least one replicate sample) **and families (bottom;** average rel. abund. > 5 % in at least one area) **and their average relative abundance (± standard deviation) per area.** Note that *Halalaimus* is highlighted in grey as this is the genus that was identified up to species level.

| Genus | APEI-3 | IFREMER | GSR | IOM | BGR_RA | BGR_PA |
|---|---|---|---|---|---|---|
| *Acantholaimus* | 7.17 ± 5.10 | 9.15 ± 3.63 | 12.77 ± 3.55 | 9.05 ± 2.91 | 17.95 ± 7.51 | 25.42 ± 0.33 |
| *Chromadorita* | 0.89 ± 0.87 | 1.44 ± 1.22 | 3.76 ± 2.46 | 1.21 ± 0.75 | 3.00 ± 1.08 | 3.04 ± 1.38 |
| *Daptonema* | 3.06 ± 2.12 | 1.17 ± 0.89 | 0.93 ± 0.90 | 1.11 ± 0.66 | 3.75 ± 0.11 | 5.46 ± 1.30 |
| *Endeolophos* | 0.58 ± 1.00 | 1.16 ± 1.75 | 0.30 ± 0.52 | 0.14 ± 0.24 | 1.13 ± 0.86 | 2.72 ± 2.26 |
| *Halalaimus* | 10.76 ± 1.63 | 2.38 ± 1.21 | 2.82 ± 1.61 | 4.70 ± 1.01 | 3.61 ± 1.82 | 5.80 ± 1.07 |
| *Leptolaimus* | 2.08 ± 1.97 | 1.47 ± 1.31 | 2.87 ± 1.80 | 1.63 ± 0.61 | 2.51 ± 2.36 | 2.18 ± 0.75 |
| *Microlaimus* | 1.87 ± 1.84 | 3.16 ± 2.87 | 1.89 ± 0.95 | 4.59 ± 2.59 | 1.81 ± 1.06 | 0.46 ± 0.08 |
| *Molgolaimus* | 2.81 ± 2.75 | 0.73 ± 0.91 | 1.50 ± 2.60 | 1.76 ± 1.53 | 2.75 ± 0.26 | 1.20 ± 2.08 |
| *Monhystrella* | 26.09 ± 6.55 | 31.14 ± 3.96 | 25.81 ± 6.03 | 26.45 ± 5.41 | 27.53 ± 12.13 | 21.75 ± 2.05 |
| *Prochromadorella* | 0.31 ± 0.53 | 0.00 ± 0.00 | 2.10 ± 3.64 | 0.00 ± 0.00 | 0.00 ± 0.00 | 0.00 ± 0.00 |
| *Prototricoma* | 1.16 ± 2.01 | 6.01 ± 1.70 | 4.10 ± 0.74 | 1.63 ± 0.61 | 0.00 ± 0.00 | 0.00 ± 0.00 |
| *Thalassomonhystera* | 4.29 ± 2.65 | 4.15 ± 2.77 | 2.78 ± 1.55 | 4.36 ± 1.61 | 2.45 ± 1.34 | 3.27 ± 1.72 |
| *Theristus* | 0.61 ± 1.06 | 2.35 ± 2.22 | 0.97 ± 0.99 | 4.34 ± 4.04 | 1.71 ± 1.06 | 0.68 ± 0.79 |
| *Tricoma* | 3.31 ± 2.61 | 2.18 ± 0.97 | 2.26 ± 2.33 | 2.29 ± 1.30 | 0.29 ± 0.27 | 1.56 ± 0.45 |

| Family | APEI-3 | IFREMER | GSR | IOM | BGR_RA | BGR_PA |
|---|---|---|---|---|---|---|
| **Chromadoridae** | 8.94 ± 6.69 | 12.98 ± 5.03 | 21.71 ± 7.51 | 11.45 ± 2.22 | 23.56 ± 9.55 | 32.92 ± 2.57 |
| **Desmoscolecidae** | 5.40 ± 4.78 | 10.09 ± 1.11 | 7.94 ± 3.46 | 5.31 ± 2.68 | 1.42 ± 1.03 | 2.37 ± 0.81 |
| **Diplopeltidae** | 2.50 ± 1.52 | 2.60 ± 1.54 | 4.40 ± 0.61 | 5.49 ± 3.27 | 3.85 ± 0.59 | 1.56 ± 0.47 |
| **Microlaimidae** | 2.48 ± 1.45 | 3.39 ± 2.77 | 3.14 ± 2.99 | 6.21 ± 2.91 | 2.43 ± 1.06 | 0.76 ± 0.26 |
| **Monhysteridae** | 34.69 ± 8.10 | 39.37 ± 5.53 | 29.54 ± 4.22 | 36.30 ± 3.93 | 30.61 ± 11.89 | 27.33 ± 1.33 |
| **Oxystominidae** | 12.65 ± 3.35 | 3.81 ± 1.27 | 5.37 ± 2.44 | 5.34 ± 2.05 | 4.73 ± 1.94 | 7.32 ± 0.67 |
| **Xyalidae** | 14.71 ± 1.74 | 12.87 ± 4.20 | 7.93 ± 3.75 | 16.39 ± 1.57 | 13.84 ± 1.47 | 13.51 ± 4.50 |





**Table 7. Species counts for *Halalaimus* per area.** Number of individuals per area for each species, ranked according to decreasing geographical spread. When individuals were strongly resembling a certain species but without a 100 % match for all the characteristics, they were labeled as 'affinity' (aff.) to the closest species.

| | APEI-3 | IFREMER | GSR | IOM | BGR_RA | BGR_PA |
|---|---|---|---|---|---|---|
| *Halalaimus abyssus* Bussau, 1993 | 4 | 1 | 2 | 6 | 6 | 6 |
| *Halalaimus egregius* Bussau, 1993 | 2 | 1 | 1 | 3 | 5 | 4 |
| *Halalaimus longinquus* Bussau, 1993 | 6 | 1 | 1 | 2 | 4 | 3 |
| *Halalaimus oblongus* Bussau, 1993 | 4 | 2 | 1 | 1 | 5 | 4 |
| *Halalaimus praestans* Bussau, 1993 | 2 | 2 | 1 | 3 | 5 | 2 |
| *Halalaimus absconditus* Bussau, 1993 | 2 | 1 | 1 | | 5 | 3 |
| *Halalaimus aedificandistudiosus* Bussau, 1970 | 1 | | | | 1 | 1 |
| *Halalaimus* aff. *amphidellus* Vitiello, 1970 | 1 | | | 1 | | 2 |
| *Halalaimus* aff. *delamarei* Vitiello, 1970 | | | 1 | 1 | | 4 |
| *Halalaimus* aff. *absconditus* Bussau, 1993 | 1 | | | | | 1 |
| *Halalaimus* aff. *marri* Mawson, 1958 | 1 | 1 | | | | |
| *Halalaimus* aff. *praestans* Bussau, 1993 | 1 | | | 2 | | |
| *Halalaimus sp. indeterminabilis* | 2 | | | 1 | | |
| *Halalaimus arundinaceus* Bussau, 1993 | 1 | | | 1 | | |
| *Halalaimus filicorpus* Vitiello, 1970 | 2 | | | | | 1 |
| *Halalaimus longicolis* Allgén, 1932 | 1 | | | | | 1 |
| *Halalaimus* aff. *abyssus* Bussau, 1993 | 2 | | | | | |
| *Halalaimus* aff. *egregius* Bussau, 1993 | | | | | | 3 |
| *Halalaimus* aff. *longinquus* Bussau, 1993 | | | | 1 | | |
| *Halalaimus* aff. *oblongus* Bussau, 1993 | | 1 | | | | |
| *Halalaimus* aff. *tenuicapitatus* Filipjev, 1946 | 1 | | | | | |
| *Halalaimus* aff. *turbidus* Vitiello, 1970 | | | | 1 | | |
| *Halalaimus n. sp.1* | 1 | | | | | |
| *Halalaimus n. sp.2* | | | | | | 1 |
| **Total species** | 18 | 8 | 7 | 12 | 7 | 14 |
| **Total individuals** | 35 | 10 | 8 | 23 | 31 | 36 |





**Figures**

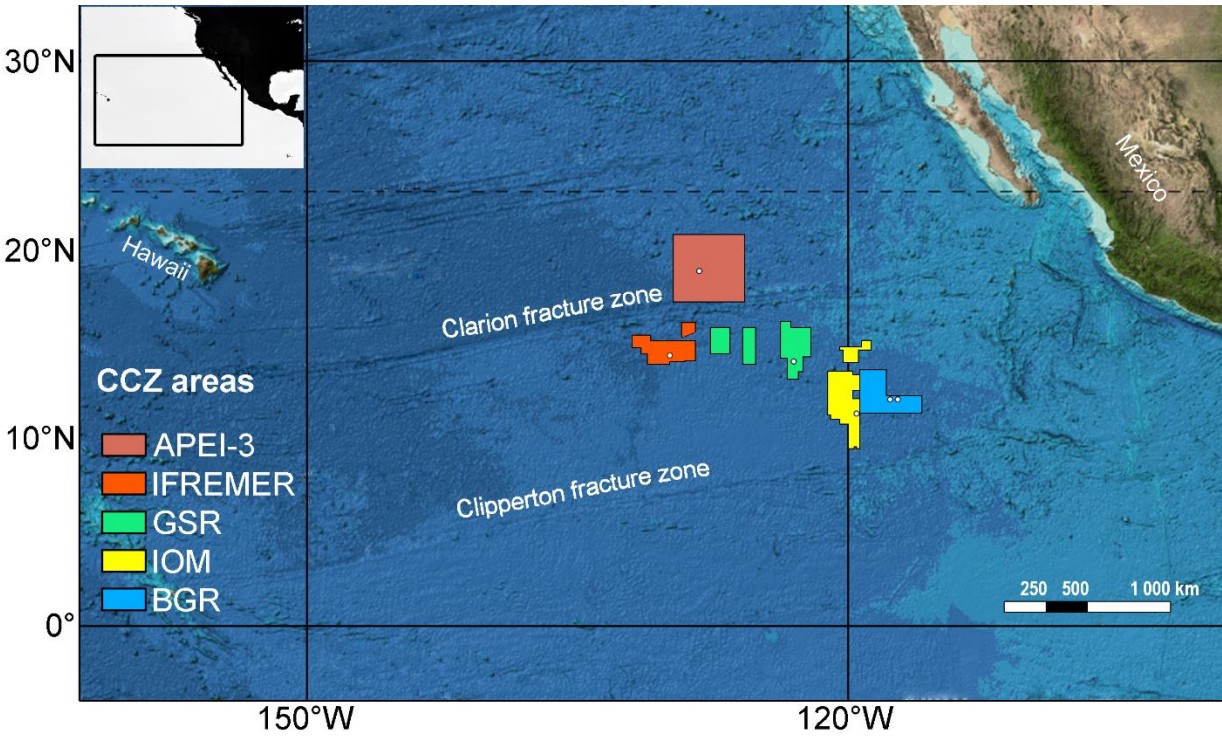

**Figure 1: Overview map of geographical sampling region and different license areas studied.** Colour code based on (Vanreusel et al., 2016) and will be maintained throughout the rest of the manuscript. Base map modified from GEBCO (www.gebco.net). Colour gradient from light to dark represents bathymetry.



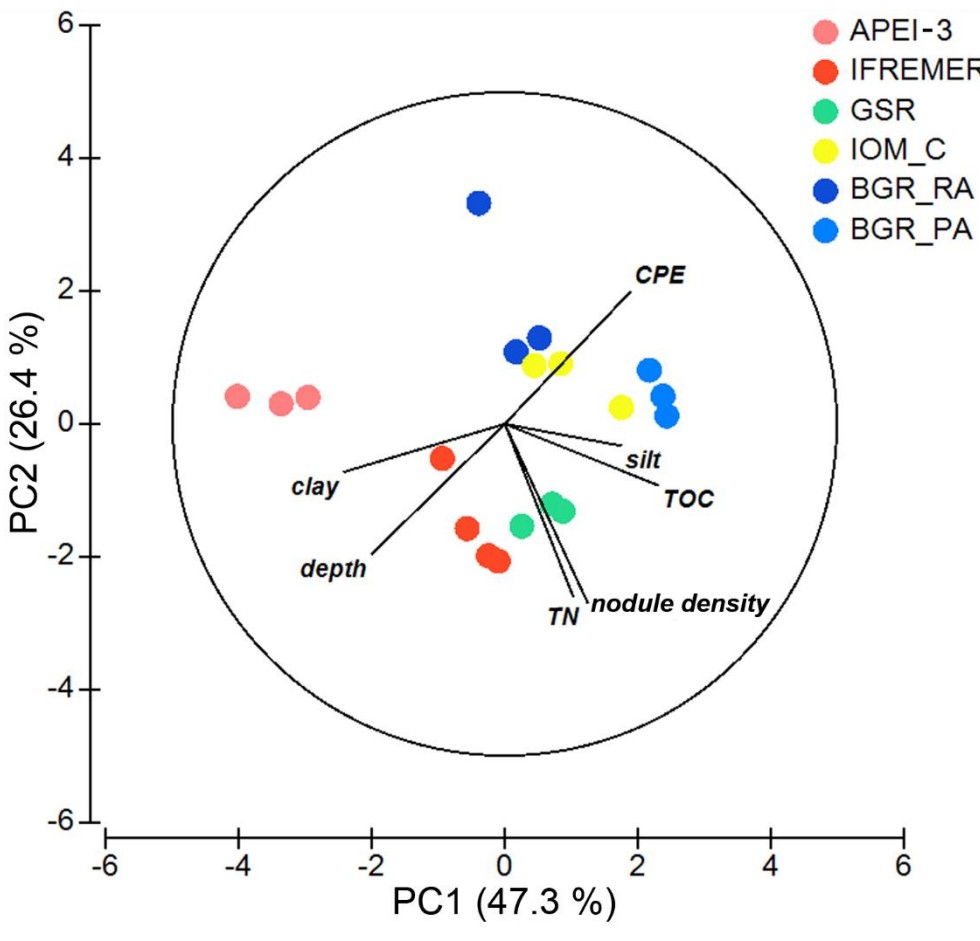

**Figure 2. PCA plot of the different areas according to environmental conditions.** All variables were normalized prior to analysis. Note that silt and clay content was log-transformed to account for skewness in the data. Numbers along PCA axes represent the total variation percentage explained by that axis.



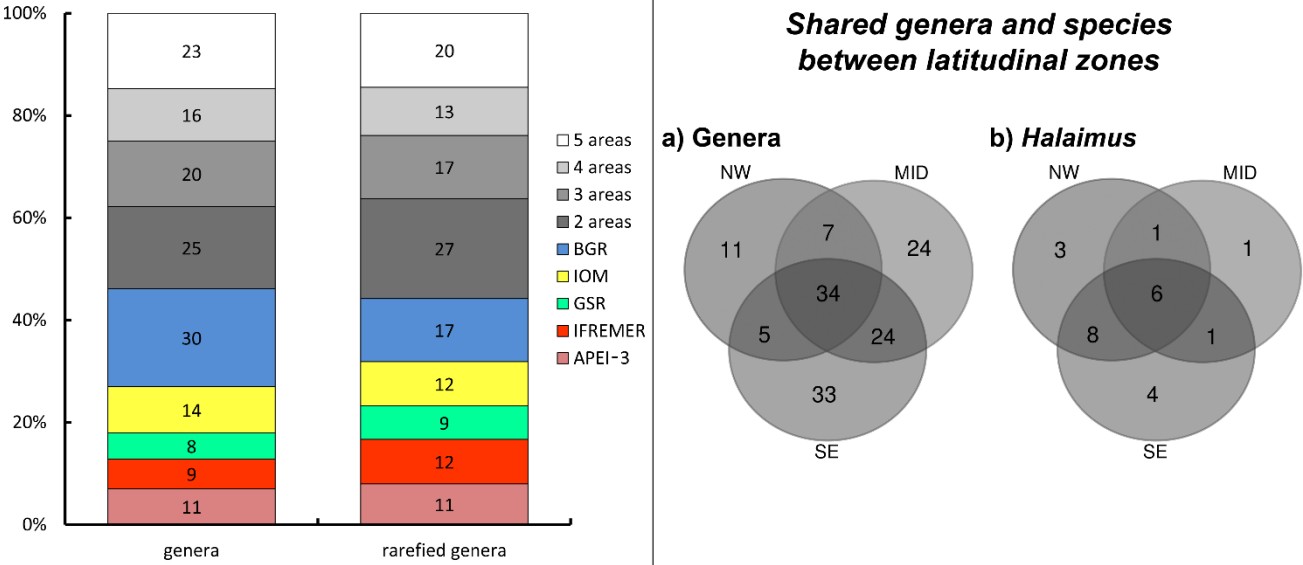

**Figure 3. Geographical distribution of genera and *Halalaimus* species.** Left panel: Number of shared genera between the different areas, and their relative contribution to totals, based on original count data, as well as rarefied genus numbers. Note that BGR_RA and BGR_PA were considered as 1 area in this case. Values along the Y-axis represent the relative contribution of unique and shared genera, while the number indicated in each bar represents the absolute count. Right panel: a) partitioning of genera between latitudinal zones (NW = APEI-3; MID = IFREMER + GSR; SE = IOM + BGR). Genus numbers based on rarefied counts. b) partitioning of *Halalaimus* species between latitudinal zones.