# Peer review of "Distribution of free-living marine nematodes in the Clarion-Clipperton Zone: implications for future deep-sea mining scenarios"

_Biogeosciences, 2018_

## Referee Comment (RC1) · Anonymous Referee #1 · 3 Feb 2019

This manuscript presented the diversity of nematodes in surface 5 cm of sediments in the license areas (of Germany, IOM, Belgium and France) and reference area of the CCZ. The data are important to help make a decision on the mining method and designation of source areas to protect the seabed ecosystem when large scale mining of the nodules is carried out.

In the part of Materials and Methods, the authors stated that the IOM samples were sliced per cm down to 5 cm sediment depth. However, there is no mention of this result. Please clarify that authors only used the total amount instead of different depth.

[Figure]

BGR was further divided into BGR_RA and BGR_PA, but these subareas were not labelled in Fig. 1. Please try to label them so that readers could know the sampling sites.

IOM should be the Interoceanmetal Joint Organization https://iom.gov.pl/. In the case of this manuscript with the same background of seabed mining, this abbreviation should be occupied by this organization. Therefore, I recommend the authors to have another abbreviation for Inter-Ocean Metal consortium (or maybe they are the same one in different forms).

The title is "geographic distribution. . .". Since all the sites are in the CCZ with similar geographical characteristics, I recommend the title to be "distribution. . .".

———————————————————

---

## Referee Comment (RC2) · Anonymous Referee #2 · 8 Apr 2019

This MS is on deep-sea meiofaunal distribution and diversity. Dataset is rather small, and analyses are simple. But the studied site CCZ is well-known deep-sea nodule mining area proposed and solid environmental/ecological assessment is urgently needed. In this regard, the dataset and analyses are basically reasonable enough. So the MS is of high importance and timely. I suggest relatively minor revision. Broader citation on deep-sea ecology & biodiversity and their environmental control is needed (as detailed below).

Specific comments are listed below:

[Figure]

P. 2, Line 11 ". . .poorly understood (. . ." Add Sweetman et al. and Yasuhara & Danovaro 2016

Sweetman, A. K., Thurber, A. R., Smith, C. R., Levin, L. A., Mora, C., Wei, C. L., Gooday, A. J., Jones, D. O. B., Rex, M., Yasuhara M., Ingels, J., Ruhl, H. A., Frieder, C. A., Danovaro, R., Würzberg, L., Baco, A., Grupe, B. M., Pasulka, A., Meyer, K. S., Dunlop, K. M., Henry, L. A., Roberts, J. M., 2017. Major impacts of climate change on deep-sea benthic ecosystems. Elementa: 5, doi:10.1525/elementa.203.

Yasuhara, M. and Danovaro, R., 2016. Temperature impacts on deep-sea biodiversity. Biological Reviews, 91: 275–287.

P. 2, Line 14 ". . .complicated by low sedimentation rates at these depths" Why?

P. 2, Line 15 ". . .the smaller-sized macro- and meiofauna (Ramirez-Llodra et al., 2010)." Add classic Hessler and Sanders papers for high deep-sea biodiversity.

P. 2, Line 15-17 "This is partially a consequence of high heterogeneity that exists as a result of the complex geological and hydrological features of the deep seafloor (Vanreusel et al., 2010), which create microscale patchiness in both abiotic and biotic features." It seems the authors' result doesn't support this statement. Dominant species are the same in all the site, showing low heterogeneity. You may site Levin et al reviews and some Yasuhara and McClain papers on temperature vs POC issues. Levin et al., 2001. Ann. Rev. Earth Planet. Sci. 32, 51–93

Yasuhara, M., Hunt, G., Cronin, T.M. and Okahashi, H., 2009. Temporal latitudinal-gradient dynamics and tropical instability of deep-sea species diversity. Proceedings of the National Academy of Sciences of the United States of America, 106: 21717–21720.

Yasuhara, M., Okahashi, H., Cronin, T.M., Rasmussen, T.L. and Hunt, G., 2014. Response of deep-sea biodiversity to abrupt deglacial and Holocene climate changes in the North Atlantic Ocean. Global Ecology and Biogeography, 23 (9): 957–967.

McClain et al. 2012. Energetics of life on the deep seafloor. PNAS

Tittensor et al. 2011. Species–energy relationships in deep-sea molluscs. Biology Letters

P. 3, Line 29 and other places. Brackets usage are strange in several places like "using the guide of (Higgins and Thiel, 1988)."

P. 4, Line 12 "For this study, clay (sediment particles < 4 $\mu$m) and silt (4–63 $\mu$m) size fractions were considered in further analyses." Good to specify what are the "further analyses"

Did you address spatial auto-correlation for your regression models?

P. 7 line 1. "repeatedly (e.g., (Wei et al., 2010))." Check the house style of bracket usage. Not only here, but also other places.

P. 9, line 4. "only few genera occurred in higher numbers" Higher than what? This sentence is unclear a bit.

P. 9 Line 30: "...ecosystem functioning" Cite Danovaro and Yasuhara papers regarding ecosystem functioning.

Yasuhara, M., Doi, H., Wei, C. L., Danovaro, R. and Myhre, S. E., 2016. Biodiversity–ecosystem functioning relationships in long-term time series and palaeoecological records: deep sea as a test bed. Philosophical Transactions of the Royal Society B: doi:10.1098/rstb.2015.0282

Danovaro et al., 2008. Exponential Decline of Deep-Sea Ecosystem Functioning Linked to Benthic Biodiversity Loss. Current Biology

In Discussion, the authors emphasized high number/diversity of rare genera and species. But such is not emphasized in the abstract. These rare taxa tend to be found in only one site and thus are higjly vulnerable for mining, right? If this is correct, the authors need to emphasize this more.

Discussion is a bit lengthy too much and can be simpler, given the simple and small dataset. It seems the authors talk too much based on small data.

Do you see any environmental control of diversity in your data?

–End of Letter–

---

## Author Comment (AC1) · 17 Jun 2019

Author's response to Referee's comments (RC1)

RC1: This manuscript presented the diversity of nematodes in surface 5 cm of sediments in the license areas (of Germany, IOM, Belgium and France) and reference area of the CCZ. The data are important to help make a decision on the mining method and designation of source areas to protect the seabed ecosystem when large scale mining of the nodules is carried out.

Referee comment: In the part of Materials and Methods, the authors stated that the IOM samples were sliced per cm down to 5 cm sediment depth. However, there is no mention of this result. Please clarify that authors only used the total amount instead of different depth.

Author's response: This is indeed a valuable comment by the reviewer, and we failed to mention this properly in the first version of the manuscript. Therefore, we have added a sentence to the respective paragraph in the Material and Methods section. It now reads (Material and Methods, lines 30-32): "From each cm-slice (IOM) or bulk sample (all other areas), between 120 and 320 nematodes were randomly picked, transferred to anhydrous glycerol (Seinhorst, 1959;De Grisse, 1969) and mounted on slides. Later on, genus counts for the different sediment layers of the IOM samples were summed prior to statistical analysis."

Referee comment: BGR was further divided into BGR_RA and BGR_PA, but these subareas were not labelled in Fig. 1. Please try to label them so that readers could know the sampling sites.

Author's response: The Figure 1 has been adapted and now includes labels for BGR_RA and BGR_PA. Also the Figure caption has been updated so that it is clear which sampling locations were investigated.

Referee comment: IOM should be the Interoceanmetal Joint Organization https://iom.gov.pl/. In the case of this manuscript with the same background of seabed mining, this abbreviation should be occupied by this organization. Therefore, I recommend the authors to have another abbreviation for Inter-Ocean Metal consortium (or maybe they are the same one in different forms).

Author's response: The reviewer is correct to assume that we mean the same organization and that the abbreviation should be Interoceanmetal Joint Organization. This has been adapted in the respective paragraph in the Material and Methods section.

Referee comment: The title is "geographic distribution: ". Since all the sites are in the CCZ with similar geographical characteristics, I recommend the title to be "distribution: ".

Author's response: Although we do believe that samples for this study were collected over a rather large geographic range, we do see the point that the reviewer is making here. So we are willing to remove the word 'Geographic' from the title of the manuscript since it would not change the focus of it.

[Figure]

**Fig. 1.** Figure 1: Overview map of geographical sampling region and different license areas studied.

---

## Author Comment (AC2) · 23 Jun 2019

Author's response to Referee's comments (RC2)

RC2: This MS is on deep-sea meiofaunal distribution and diversity. Dataset is rather small, and analyses are simple. But the studied site CCZ is well-known deep-sea nodule mining area proposed and solid environmental/ecological assessment is urgently needed. In this regard, the dataset and analyses are basically reasonable enough. So the MS is of high importance and timely. I suggest relatively minor revision. Broader

citation on deep-sea ecology & biodiversity and their environmental control is needed (as detailed below).

Specific comments are listed below:

Referee comment: P. 2, Line 11 "...poorly understood (..." Add Sweetman et al. and Yasuhara & Danovaro 2016

• Sweetman, A. K., Thurber, A. R., Smith, C. R., Levin, L. A., Mora, C., Wei, C. L., Gooday, A. J., Jones, D. O. B., Rex, M., Yasuhara M., Ingels, J., Ruhl, H. A., Frieder, C. A., Danovaro, R., Würzberg, L., Baco, A., Grupe, B. M., Pasulka, A., Meyer, K. S., Dunlop, K. M., Henry, L. A., Roberts, J. M., 2017. Major impacts of climate change on deep-sea benthic ecosystems. Elementa: 5, doi:10.1525/elementa.203. • Yasuhara, M. and Danovaro, R., 2016. Temperature impacts on deep-sea biodiversity. Biological Reviews, 91: 275–287.

Author's response: We have consulted the proposed publications and have made proper reference to them where deemed appropriate. Although their focus is more on climate change impacts, which is not the main topic of our manuscript, they also touch upon the mining-related issues (especially Sweetman et al., 2017).

Referee comment: P. 2, Line 14 "...complicated by low sedimentation rates at these depths" Why?

Author's response: With this sentence we mean that the phytoplankton-derived food input into the benthic system is typically rather low at abyssal depths (see also the proposed Sweetman et al. 2017 paper from the previous comment). Especially in this area, we know that primary production is small and sedimentation of phytodetritus low. We have added this link with the food input in the sentence which now reads (P. 2, lines 13-16): "Despite this dependency, which is further complicated by low sedimentation rates hence low food input at these depths (Petersen et al., 2016;Gollner et al., 2017;Sweetman et al., 2017), biodiversity is high in most deep-sea habitats, especially

for the smaller-sized macro- and meiofauna (Ramirez-Llodra et al., 2010;Hessler and Sanders, 1967)."

Referee comment: P. 2, Line 15 "...the smaller-sized macro- and meiofauna (Ramirez-Llodra et al., 2010)." Add classic Hessler and Sanders papers for high deep-sea biodiversity.

Author's response: This paper (Hessler & Sanders, 1967) has been added, which focuses on the macrofauna.

Referee comment: P. 2, Line 15-17 "This is partially a consequence of high heterogeneity that exists as a result of the complex geological and hydrological features of the deep seafloor (Vanreusel et al., 2010), which create microscale patchiness in both abiotic and biotic features." It seems the authors' result doesn't support this statement. Dominant species are the same in all the site, showing low heterogeneity. You may site Levin et al reviews and some Yasuhara and McClain papers on temperature vs POC issues. • Levin et al., 2001. Ann. Rev. Earth Planet. Sci. 32, 51–93 • Yasuhara, M., Hunt, G., Cronin, T.M. and Okahashi, H., 2009. Temporal latitudinal gradient dynamics and tropical instability of deep-sea species diversity. Proceedings of the National Academy of Sciences of the United States of America, 106: 21717– 21720. • Yasuhara, M., Okahashi, H., Cronin, T.M., Rasmussen, T.L. and Hunt, G., 2014. Response of deep-sea biodiversity to abrupt deglacial and Holocene climate changes in the North Atlantic Ocean. Global Ecology and Biogeography, 23 (9): 957–967. • McClain et al. 2012. Energetics of life on the deep seafloor. PNAS • Tittensor et al. 2011. Species–energy relationships in deep-sea molluscs. Biology Letters

Author's response: We partly agree with the reviewer that our results don't seem to support the statement as this is indeed true for the dominant taxa. Nevertheless, the more rare taxa do differ a lot among locations, which suggests that at least part of the nematode community varies substantially over the spatial scale under consideration in this MS. We have consulted the proposed literature and have made reference to these

publications where deemed appropriate.

Referee comment: P. 3, Line 29 and other places. Brackets usage are strange in several places like "using the guide of (Higgins and Thiel, 1988)."

Author's response: We are aware of the strange use of brackets, which is related to the EndNote insertion function. Although we adopted the required house style, the inline references still appear in this format. We will adapt this together with the editorial team once the paper gets accepted for publication.

Referee comment: P. 4, Line 12 "For this study, clay (sediment particles < 4 $\mu$m) and silt (4–63 $\mu$m) size fractions were considered in further analyses." Good to specify what are the "further analyses" Did you address spatial auto-correlation for your regression models?

Author's response: We have added the specifications for the further analyses and the sentence now reads: "For this study, clay (sediment particles < 4 $\mu$m) and silt (4–63 $\mu$m) size fractions were considered in further analyses linking nematode communities to environmental parameters." As for the spatial auto-correlation, we did check for multicollinearity in the environmental variables at the start of building the regression models, and this statement has now been added in the Material and Methods section. However, we did not specifically check for spatial auto-correlation using other methods since we believe that this would over-complicate the analyses. As pointed out by the reviewer as well, we are dealing with a limited number of sampling locations, at relatively large geographic distances, and we do not aim to generalize our findings to other unsampled locations. From previous studies, we know that nematode communities tend to differ even at very small spatial scales (e.g., (Gallucci et al., 2009;Fonseca et al., 2010;Leduc et al., 2012), so that dissimilarity between communities does not necessarily increase with increasing distance among sampling locations (see also (Hauquier et al., 2018)). Nevertheless, we are aware of the potential bias in our dataset since our different sampling locations do differ in environmental setting and other –potentially

important- environmental variables which may structure community composition and abundance have not been measured. This means that there is per definition a spatial and latitudinal gradient underlying our environmental differences as well as our community differences.

Referee comment: P. 7 line 1. "repeatedly (e.g., (Wei et al., 2010))." Check the house style of bracket usage. Not only here, but also other places.

Author's response: See reply to an earlier comment.

Referee comment: P. 9, line 4. "only few genera occurred in higher numbers" Higher than what? This sentence is unclear a bit.

Author's response: We agree that this sentence is not clear enough and have added the cut-off percentage which we considered as 'higher numbers'. The sentence now reads (P. 9, lines 7-9): "With a total of 156 genera reported in this study, genus diversity was indeed substantial on a regional scale and only few genera occurred in higher numbers (i.e. relative abundance > 5 %; Tables 5, 6)."

Referee comment: P. 9 Line 30: "...ecosystem functioning" Cite Danovaro and Yasuhara papers regarding ecosystem functioning.

• Yasuhara, M., Doi, H., Wei, C. L., Danovaro, R. and Myhre, S. E., 2016. Biodiversity– ecosystem functioning relationships in long-term time series and palaeoecological records: deep sea as a test bed. Philosophical Transactions of the Royal Society B: doi:10.1098/rstb.2015.0282 • Danovaro et al., 2008. Exponential Decline of Deep-Sea Ecosystem Functioning Linked to Benthic Biodiversity Loss. Current Biology

Author's response: We have added these references where deemed appropriate.

Referee comment: In Discussion, the authors emphasized high number/diversity of rare genera and species. But such is not emphasized in the abstract. These rare taxa tend to be found in only one site and thus are highly vulnerable for mining, right? If this is correct, the authors need to emphasize this more.

Author's response: We agree with the reviewer and have adapted the abstract accordingly. The last paragraph now reads: "Nematodes were the most abundant meiobenthic taxon and their assemblages were typically dominated by a few genera (generally 2–6) accounting for 40–70 % of all individuals, which were also widely spread along the CCZ and shared among all sampled license areas. However, almost half of the communities consisted of rare genera each contributing less than 5 % to the overall abundances and displaying a distribution which was usually restricted to a single license area. The same observations (dominant and widely spread versus rare and scattered) could be made for the species of one of the dominant genera, Halalaimus, implying that it might be mainly these rare genera and species that will be vulnerable to mining-induced changes in their habitat."

Referee comment: Discussion is a bit lengthy too much and can be simpler, given the simple and small dataset. It seems the authors talk too much based on small data. Do you see any environmental control of diversity in your data?

Author's response: Although we understand the reviewer's comment on the small dataset and rather lengthy discussion, we do believe that we should touch upon all these topics (without making too strong of a statement) in the light of future mining plans. Unfortunately, sampling at these depths and in these regions is always prone to logistic challenges, which makes the acquisition of extensive datasets very difficult. Nevertheless, even with a small dataset, we can at least hint at the potential impacts that mining operations may have and the vulnerability of the system as such. As monitoring programs are being established and data continues to being gathered in the respective license areas, information contained in this paper can help serving as a first baseline and new insights can either confirm or modify our conclusions made here. We therefore have made only a few modifications to shorten certain parts of the discussion.

Since the number of genera and diversity in general is relatively similar for the different sampling locations, we did not specifically check for a link with the environmental parameters of the sites. Based on other analyses that were not included in the manuscript

(since they did not contribute any new insights), we also did not see any patterns in the respective proportions of different nematode feeding types (deposit feeders, predators, etc.). If there would be a clear environmental control, we would have expected that, for instance, the proportion of deposit feeders would be higher in those areas where organic matter content in the sediments is higher as well but this was not the case.

Fonseca, G., Soltwedel, T., Vanreusel, A., and Lindegarth, M.: Variation in nematode assemblages over multiple spatial scales and environmental conditions in Arctic deep seas, Progress in Oceanography, 84, 174-184, 10.1016/j.pocean.2009.11.001, 2010. Gallucci, F., Moens, T., and Fonseca, G.: Small-scale spatial patterns of meiobenthos in the Arctic deep sea, Marine Biodiversity, 39, 9-25, 10.1007/s12526-009-0003-x, 2009. Gollner, S., Kaiser, S., Menzel, L., Jones, D. O. B., Brown, A., Mestre, N. C., van Oevelen, D., Menot, L., Colaço, A., Canals, M., Cuvelier, D., Durden, J. M., Gebruk, A., Egho, G. A., Haeckel, M., Marcon, Y., Mevenkamp, L., Morato, T., Pham, C. K., Purser, A., Sanchez-Vidal, A., Vanreusel, A., Vink, A., and Martinez Arbizu, P.: Resilience of benthic deep-sea fauna to mining activities, Marine Environmental Research, 129, 76-101, https://doi.org/10.1016/j.marenvres.2017.04.010, 2017. Hauquier, F., Verleyen, E., Tytgat, B., and Vanreusel, A.: Regional-scale drivers of marine nematode distribution in Southern Ocean continental shelf sediments, Progress in Oceanography, 165, 1-10, https://doi.org/10.1016/j.pocean.2018.04.005, 2018. Hessler, R. R., and Sanders, H. L.: Faunal diversity in the deep-sea, Deep-Sea Research, 14, 65-78, 1967. Leduc, D., Rowden, A. A., Bowden, D. A., Nodder, S. D., Probert, P. K., Pilditch, C. A., Duineveld, G. C. A., and Witbaard, R.: Nematode beta diversity on the continental slope of New Zealand: spatial patterns and environmental drivers, Marine Ecology Progress Series, 454, 37-52, 2012. Petersen, S., Krätschell, A., Augustin, N., Jamieson, J., Hein, J. R., and Hannington, M. D.: News from the seabed: Geological characteristics and resource potential of deep-sea mineral resources, Marine Policy, 70, 175-187, https://doi.org/10.1016/j.marpol.2016.03.012, 2016. Ramirez-Llodra, E., Brandt, A., Danovaro, R., De Mol, B., Escobar, E., German, C. R., Levin, L. A., Martinez Arbizu, P., Menot, L., Buhl-Mortensen, P., Narayanaswamy, B. E., Smith, C. R., Tittensor, D.

P., Tyler, P. A., Vanreusel, A., and Vecchione, M.: Deep, diverse and definitely different: unique attributes of the world's largest ecosystem, Biogeosciences, 7, 2851-2899, 10.5194/bg-7-2851-2010, 2010. Sweetman, A., Thurber, A., Smith, C., Levin, L., Mora, C., Wei, C., Gooday, A., Jones, D., Rex, M., Yasuhara, M., Ingels, J., Ruhl, H., Frieder, C., Danovaro, R., Würzberg, L., Baco, A., Grupe, B., Pasulka, A., Meyer, K., Dunlop, K., Henry, L.-A., and Roberts, J.: Major impacts of climate change on deep-sea benthic ecosystems, Elementa Science of the Anthropocene, doi.org/10.1525/elementa.203, 2017.